# Combining two genetic sexing strains allows sorting of non-transgenic males for *Aedes* genetic control

Célia Lutrat [1,2,3,4✉], Myriam Burckbuchler[4], Roenick Proveti Olmo [4], Rémy Beugnon [5,6], Albin Fontaine[7], Omar S. Akbari [8], Rafael Argilés-Herrero[9], Thierry Baldet [1,10], Jérémy Bouyer [1,11,12,13] & Eric Marois [4,13✉]

Chemical control of disease vectoring mosquitoes *Aedes albopictus* and *Aedes aegypti* is costly, unsustainable, and increasingly ineffective due to the spread of insecticide resistance. The Sterile Insect Technique is a valuable alternative but is limited by slow, error-prone, and wasteful sex-separation methods. Here, we present four Genetic Sexing Strains (two for each *Aedes* species) based on fluorescence markers linked to the m and M sex loci, allowing for the isolation of transgenic males. Furthermore, we demonstrate how combining these sexing strains enables the production of non-transgenic males. In a mass-rearing facility, 100,000 first instar male larvae could be sorted in under 1.5 h with an estimated 0.01–0.1% female contamination on a single machine. Cost-efficiency analyses revealed that using these strains could result in important savings while setting up and running a mass-rearing facility. Altogether, these Genetic Sexing Strains should enable a major upscaling in control programmes against these important vectors.

[1] CIRAD, UMR ASTRE, F-34398 Montpellier, France. [2] ASTRE, CIRAD, INRA, Univ. Montpellier, Montpellier, France. [3] Université de Montpellier, Montpellier, France. [4] CNRS UPR9022, INSERM U1257, Université de Strasbourg, Strasbourg, France. [5] German Centre for Integrative Biodiversity Research (iDiv) Halle-Jena-Leipzig, Puschstrasse 4, 04103 Leipzig, Germany. [6] Institute of Biology, Leipzig University, Puschstrasse 4, 04103 Leipzig, Germany. [7] Unité Parasitologie et Entomologie, Département Microbiologie et maladies infectieuses, Institut de Recherche Biomédicale des Armées (IRBA), Marseille, France. [8] School of Biological Sciences, Department of Cell and Developmental Biology, University of California, San Diego, CA 92093, USA. [9] Cicindela Ltd, Valencia, Spain. [10] CIRAD, UMR ASTRE, Sainte-Clotilde, F-97490 Reunion, France. [11] CIRAD, UMR ASTRE, Saint-Pierre, F-97410 Reunion, France. [12] Insect Pest Control Sub-Programme, Joint FAO/IAEA Division of Nuclear Techniques in Food and Agriculture, International Atomic Energy Agency (IAEA), Vienna, Austria. [13] These authors jointly supervised this work: Jérémy Bouyer, Eric Marois. ✉email: celia.lutrat@outlook.com; e.marois@unistra.fr

*A*edes aegypti and *Aedes albopictus* are invasive mosquito species responsible for the transmission of many pathogens including dengue (DENV), chikungunya (CHIKV), Zika (ZIKV) and yellow fever (YFV) viruses[1,2]. Driven by climate change and worldwide trade, both vectors are spreading rapidly and it is predicted that 49% of the world population will be at risk of *Aedes*-borne diseases by 2050 in the absence of effective control measures[3,4].

Suppression of mosquito populations using genetic control is one of the most effective, sustainable and environmentally friendly alternatives to insecticide use. It relies on repeated mass releases of non-biting male mosquitoes – either sterile (the Sterile Insect Technique, SIT[5], and its derivatives including pgSIT[6]), *Wolbachia* infected (the Incompatible Insect Technique, IIT[7–9]), both[10,11], or carrying a lethal transgene (Release of Insects carrying a Dominant Lethal, RIDL)[12]. For all these interventions, an efficient sex separation method is required. Currently, *Aedes* mosquitoes are sexed based on natural pupal size dimorphism using a Hoch's sorter, which can be either fully manual[13,14] or partly automated[11]. This method, which requires homogenous pupal size and therefore density optimised larval rearing conditions, suffers from female contamination rates between 0.8 and 1% and high daily and user-to-user variability[15]. Notably, a multi-step pupal and adult sorter was recently described which allowed $1.13 \times 10^{-7}$% female contamination[9]. However, this system like all other existing methods share the drawback of sorting at a late stage and recovering less than half of the reared males[14], meaning that >75% of total pupae are reared and fed in vain.

In *Anopheles* mosquitoes, transgenic genetic sexing strains (GSSs) allowing automated sex separation of young larvae have been described[16–19]. Fluorescent markers display male-specific expression, either by using male-specific regulatory sequences or by linking markers to the Y chromosome. One *Anopheles coluzzii* sexing strain is X-linked[20], rendering females more fluorescent than males. Sex separation is accomplished using a Complex Object Parametric Analyzer and Sorter (COPAS) device, which functions as a flow cytometer, sorting large particles according to their fluorescence. Additionally, as an alternative to the release of transgenic males, a crossing scheme generating non-transgenic male-only populations using COPAS was proposed[21]. This scheme requires a strain carrying a fluorescence marker on the Y chromosome and another carrying a fluorescence marker on the X chromosome. Crossing non-transgenic (X−/X−) females from the first strain to transgenic (X+/Y−) males from the second results in progeny consisting of transgenic (X+/X−) females and non-transgenic (X−/Y−) males.

In *Aedes* mosquitoes, GSSs allowing sex-sorting of transgenic males exist. Two studies have proposed the use of a repressible female-specific flightless phenotype[22,23]. In this system, transgenic males and females are released together but adult females do not survive long as they are unable to fly. A method combining such sex-sorting and sterilization has been developed and adapted to *Ae. aegypti* for SIT[6,24]. However, this system requires perfect sex-sorting of the parental strains to be crossed, otherwise fertile transgenic mosquitoes (some of them homozygous for a *Cas9* transgene) could be released. In *Ae. aegypti*, a red-eye GSS has also been developed and proven efficient for sorting at the pupal stage but requires automation[25]. Recently, we developed a GSS for *Ae. albopictus* carrying a fluorescence marker within a masculinizing transgenic cassette, allowing automated separation of transgenic males from non-transgenic females[26].

Here, we examined if a crossing scheme similar to the one for *Anopheles* could be designed in *Aedes* mosquitoes despite the absence of heteromorphic sex chromosomes. In *Aedes*, sex is encoded by non-homologous sex loci located on the first pair of autosomes. The male locus is called "M", while the common locus

is called "m"; thus, females are m/m and males are m/M. These sex loci are about 1.18 Mbp long in *Ae. aegypti*[27], and are delimited by antagonistic factors that protect them from recombination[28,29]. They are embedded within a 63 Mbp region with strong male-female genetic differentiation, in which recombination remains suppressed[30]. Consequently, linking a fluorescence marker transgene to the *Aedes* sex loci or to the surrounding non-recombining region might allow automated sorting of either transgenic males directly from an M-linked strain, or of non-transgenic males after crossing hemizygous m-transgenic males with wild-type females. In this work, we developed two GSSs for both *Aedes* vector species, one linked to the M-locus and another to the m-locus, using genome editing and transgenesis. We show that these GSSs can be used separately to release transgenic mosquitoes, or combined to purify large populations of wild-type males. We discuss the suitability of the method for mass-rearing and inundative releases.

## Results

**Obtaining four genetic sexing strains.** In *Ae. aegypti*, linkage of an *eGFP* marker transgene to the m and M loci was achieved by CRISPR-Cas9 knock-in targeting a *mucin-3A* gene, AAEL019619, that was predicted to be central to the non-recombining region encompassing the sex-loci by Fontaine and colleagues[30] (Fig. 1a, b, see Methods for further details). We isolated an *Ae. aegypti* M-linked strain that we termed Aaeg-M, and an *Ae. aegypti* m-linked strain that we termed Aaeg-m. *GFP* fluorescence expressed from a ubiquitous promoter allows sex separation at the first larval stage: in Aaeg-M, males express *GFP* while females are non-transgenic (Fig. 1a) and in Aaeg-m, males express one copy of the *GFP* transgene, while females express two copies resulting in brighter fluorescence (Fig. 1b). Both lines were backcrossed 7 times to a Brazilian genetic background (Bra). In *Ae. albopictus*, M-linkage of fluorescence markers was achieved by *piggyBac* preferential insertions near the masculinization gene, *Nix*, stimulated by the inclusion of *Nix*-derived sequences in the *piggyBac* transgenesis plasmid (Fig. 1c, see Methods). A similar phenomenon, termed transposon homing, has been previously observed for P elements in *Drosophila*[31]. This approach yielded at least eight lines with tight M-linkage out of about 60 screened *piggyBac* insertions (strikingly, no or weak M-linkage was obtained with these *piggyBac* constructs devoid of *Nix* sequence). Among the tightly M-linked *Ae. albopictus* strains, we selected one expressing YFP, that we termed Aal-M. Sequencing (see "Targeted sequencing method" in Methods section) revealed that the transposon had landed in a non-coding sequence in scaffold 16, located on 1q12 according to the latest genome assembly[32]. To obtain a second sexing strain with m linkage in the absence of knowledge of the *Ae. albopictus* m locus sequence, we screened >120 *piggyBac* random insertions (Fig. 1d). The best m-linked insertion line that we obtained showed a recombination frequency of 0.1%. Its transgenic cassette harboured a *Cas9* transgene associated with a *DsRed* fluorescence marker and was flanked by lox sites. A *Cas9* transgene being undesirable in a sexing strain, we excised it using CRE recombinase and replaced it with an *eGFP* transgene. This *Ae. albopictus* m-linked strain was termed Aal-m. Sequencing of the transposon's flanking genomic sequence indicated that it had landed into a highly repeated region; hence, its exact genomic location could not be identified but matched several loci in 1q31 (see Methods and Supplementary Data 1). Both *Ae. albopictus* transgenic lines show a clear sex-separation pattern in COPAS analyses (Fig. 1c, d). All four lines were fluorescence-sorted and screened at each generation. In the Aaeg-M line, a single recombination event was visually recorded after 15 generations (>10,000 individuals

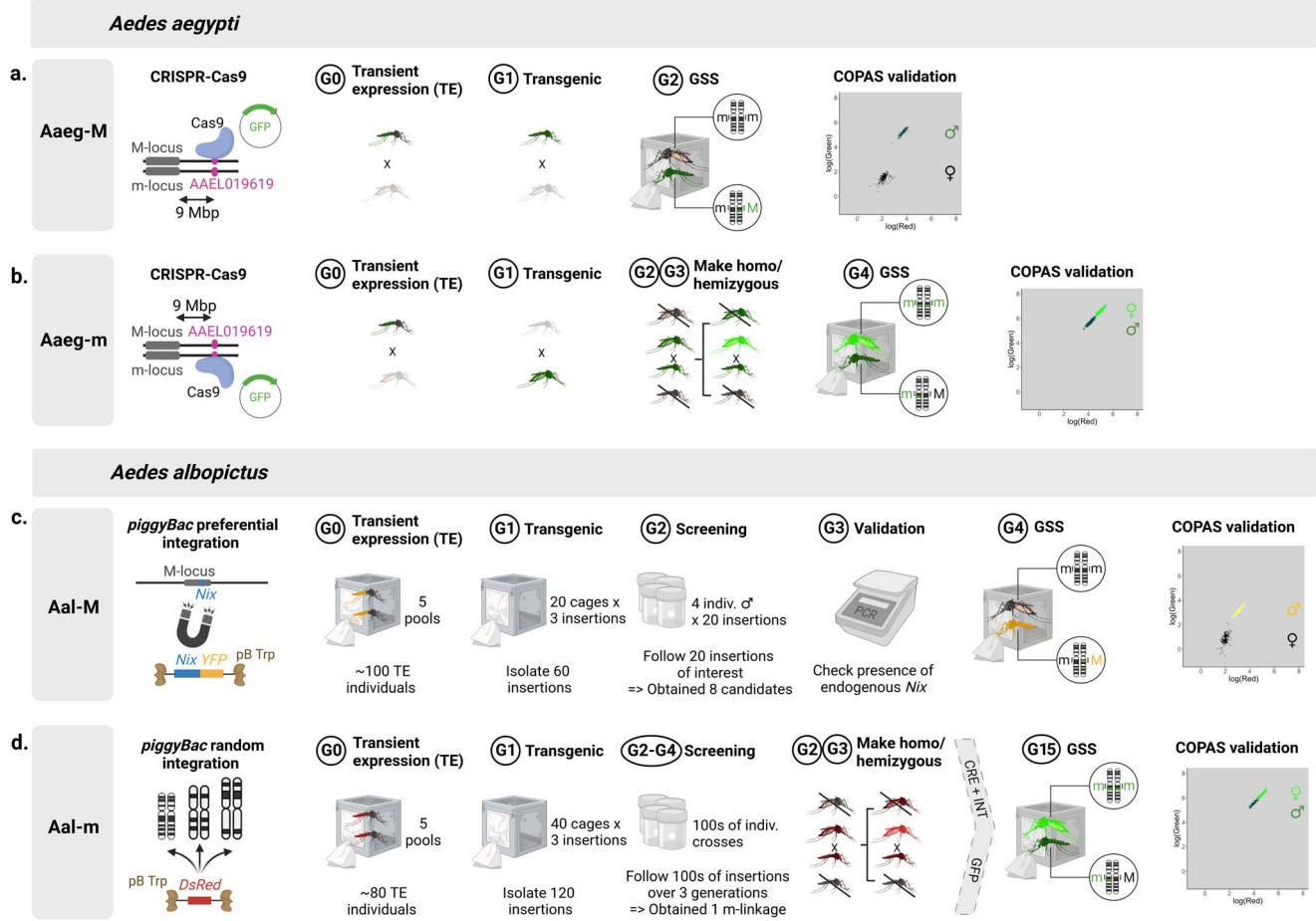

**Fig. 1 Schematic of the procedures used to obtain linkage of a fluorescence marker to the M and m-loci in *Ae. aegypti* and *Ae. albopictus*.** In *Ae. aegypti*, a single CRISPR-Cas9 target was used to obtain an M-linked strain (**a**) and an m-linked strain (**b**) within 2–4 generations. Both lines allow sex separation using a COPAS flow cytometer, as shown on the graphs generated from representative sorting data for these lines (arbitrary fluorescence units). In *Ae. albopictus*, M-linkage was obtained by *piggyBac* preferential integration near the M-locus stimulated by the presence of *Nix* sequences, requiring screening and testing for 4 generations (**c**). "pB Trp" = helper plasmid expressing piggyBac transposase. In *Ae. albopictus*, m-linkage was obtained by *piggyBac* random integration through intensive screening over 6 generations (**d**). The selected m-linked line was then modified to remove undesirable transgenes using CRE-recombinase ("CRE"), an integrase ("INT") and a *GFP* expressing plasmid docking into the attP site left-over after lox cassette excision. Both Aal-M and Aal-m allow sex separation using a COPAS flow cytometer (representative sorting from actual data shown, in COPAS arbitrary fluorescence units). Figure designed on BioRender.com.

screened in total). Consequently, we estimated that the Aaeg-M and Aaeg-m lines recombine at approximately 0.01%. In Aal-M, four recombinant individuals were observed in the tenth generation after successively screening a total of >50,000 individuals. These recombinations are likely to have arisen from a single event, suggesting that recombination is extremely rare in the Aal-M strain as well.

**Combining M and m-linked genetic sexing strains for purifying non-transgenic males.** A pilot experiment to obtain non-transgenic males using two GSS was performed in both *Ae. aegypti* and *Ae. albopictus*. In *Ae. aegypti*, 1000 negative females from Aaeg-M were COPAS-sorted (Fig. 2a) and crossed with 300 COPAS sorted hemizygous males from Aaeg-m (Fig. 2b). Their progeny (Aaeg-CS, 'CS' standing for 'Crossing Scheme') comprised 65,839 larvae, of which 32,960 were negative and supposedly males (Fig. 2c). We COPAS-extracted 2000 of those, all of which emerged as non-transgenic males. In *Ae. albopictus*, we applied the same crossing scheme using 850 negative females from the Aal-M strain crossed with 250 hemizygous males from the Aal-m strain (Fig. 2d-e). In their progeny, we obtained 24,239

Aal-CS larvae, of which 12,173 were negative and presumably males (Fig. 2f). Of the 2000 negative COPAS-extracted larvae, 1468 pupae were recovered and visually verified, revealing the presence of one contaminant non-transgenic female, which is consistent with the estimated 0.1% recombination rate of the Aal-m strain.

**Assessing the fitness of the genetic sexing strains**. The fitness of the selected M and m-linked lines, as well as of the intermediary colony from the sexing scheme (non-transgenic females from the M-linked line crossed with hemizygous transgenic males from the m-linked line), was analysed by examining the following parameters: sex ratio, fecundity (number of eggs per female), egg hatching rate, and survival from larva to adulthood. In *Ae. aegypti*, lines Aaeg-M, Aaeg-m and the intermediary colony from the crossing scheme (Aaeg-CS) showed similar sex ratios to the Bra WT line (Fig. 3a, Supplementary Table 1 and Data 2). The fecundity of Aaeg-M and Aaeg-m was not different from that of Bra (WT), while Aaeg-CS had a higher fecundity (Fig. 3b). Aaeg-M eggs hatched similarly to Bra eggs, while Aaeg-m eggs and Aaeg-CS eggs hatched significantly better (Fig. 3c). Survival from

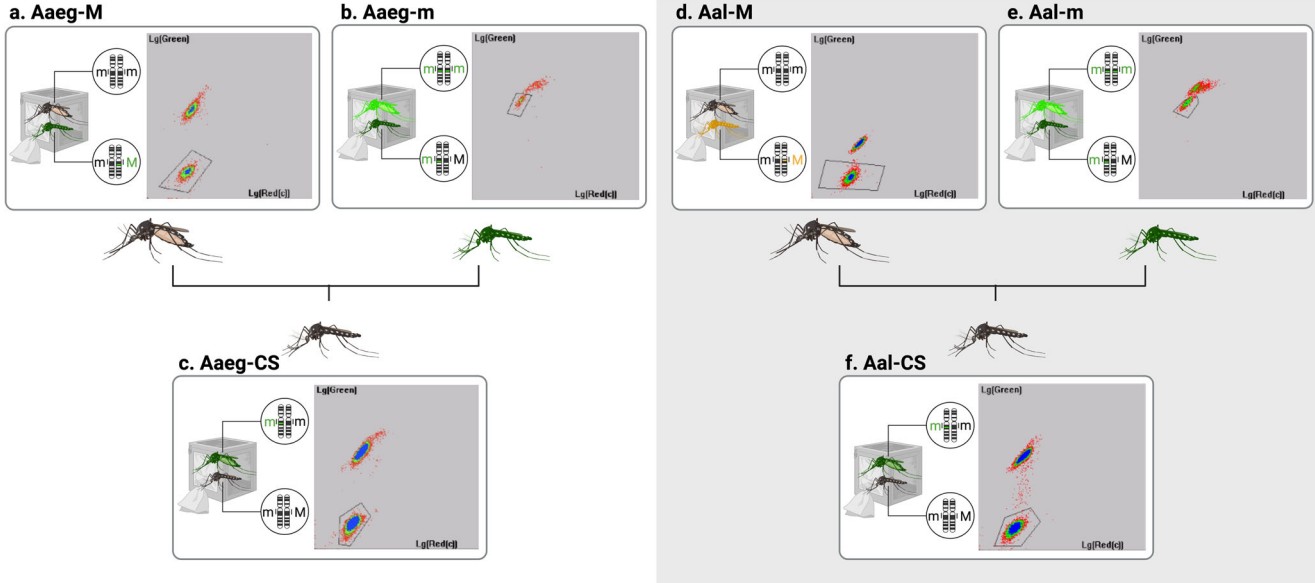

**Fig. 2 Successive sortings leading to purification of non-transgenic males.** Panels show the fluorescence graphs log(Green) = f(log(Red)) displayed by the COPAS software. No scales are displayed by the software as it uses arbitrary fluorescence units. Gated regions were defined for sorting populations of interest. Of note, both GFP and YFP fluorochromes can be read in GFP mode. **a** Sorting of non-transgenic females from Aaeg-M strain. Total larvae = 6079. **b** Sorting of hemizygous transgenic males from Aaeg-m strain. Total larvae = 711. **c** Sorting of non-transgenic males in the progeny of 1000 non-transgenic females from Aaeg-M strain crossed with 300 hemizygous transgenic males from Aaeg-m strain. Total larvae = 65,839. **d** Sorting of non-transgenic females from Aal-M strain. Total larvae = 6797. **e** Sorting of hemizygous transgenic males from Aal-m strain. Total larvae = 1960. **f** Sorting of non-transgenic males in the progeny of 850 non-transgenic females from Aal-M strain crossed with 250 hemizygous transgenic males from Aal-m strain. Total larvae = 24,239. Larvae of intermediate fluorescence were observed to be either dead fluorescent larvae or larvae with lower GFP expression at the time of the sorting. All the individuals that reached the pupal stage were confirmed to be males. Figure designed on BioRender.com.

first instar larvae to adult stages was not significantly different in Aaeg-M, Aaeg-m and Aaeg-CS compared to Bra (Fig. 3d). In *Ae. albopictus*, the sex ratios of the Aal-M, Aal-m and Aal-CS lines were not significantly different from that of the wild-type line BiA (Fig. 3e). Aal-M had significantly higher fecundity than BiA (WT), while Aal-m and Aal-CS had similar fecundities to WT (Fig. 3f). Eggs from the Aal-M and Aal-CS lines had a slightly higher hatching rate than eggs from BiA (WT), while Aal-m eggs hatched similarly (Fig. 3g). Survival from L1 larvae to adults was marginally increased in Aal-M and Aal-m compared to BiA (WT), while it was not significantly different in the progeny of the crossing scheme (Aal-CS, Fig. 3h).

**Assessing the fitness of the males to be released.** In an operational SIT approach, transgenic males from the M-linked lines or non-transgenic males from the crossing scheme would be produced for release. With this aim in mind, we assessed their fitness through competitiveness assays, flight tests and two-week survival tests under laboratory conditions. We tested the competitiveness of transgenic males by placing 30 transgenic males and 30 WT males in a cage with 30 virgin WT females and measuring the percentage of transgenic progeny in their offspring. In the competitiveness assay between transgenic and non-transgenic males in *Ae. aegypti*, Aaeg-M performed equally to Bra (WT) and Aaeg-CS (Fig. 4a, Supplementary Table 1). In *Ae. albopictus*, given the higher hatching rate and fecundity of Aal-M compared to BiA (WT) and Aal-CS, equal competitiveness would result in higher than expected percentages of Aal-M larvae in both competitiveness assays. Aal-M showed a 0.59 competitiveness compared to BiA (WT) and a similar competitiveness compared to Aal-CS (Fig. 4b), which means that both Aal-M and Aal-CS had a reduced competitiveness as compared to WT. Male flight ability was assessed through a standardized test consisting in placing 100 males in a small cup at the bottom of vertical tubes topped by a

fan and counting how many males manage to escape through the tubes[33] In flight tests, *Ae. aegypti* transgenic males and non-transgenic males from the crossing scheme performed as well as control males, while in *Ae. albopictus*, the escape rates of Aal-M males were marginally lower and those of Aal-CS males marginally higher than those of WT, with no significant difference (Fig. 4c, d). Adult survival was monitored for two weeks and no significant differences were observed between transgenic males, non-transgenic males from the crossing scheme, and males from their respective WT lines (Fig. 4e-f).

**Determining the ideal COPAS-sorting speed.** To assess the reliability of COPAS-sorting for high-throughput mass production of males in a mosquito production facility, we evaluated the contamination rate as a function of sorting speed. A pool of 10,000 Aaeg-CS first instar larvae (fluorescent females + non-fluorescent males) was repeatedly COPAS-sorted and we measured the male recovery rate and the female contamination rate at different sorting speeds. Sorting at a given speed was repeated $N = 3$ times. We observed that the percentage of recovery decreased from $91.5 \pm 1.4\%$ to $26.9 \pm 1.9\%$ when increasing the sorting speed from 6 to 300 larvae / sec, following a degree-two polynomial regression model ($R^2 = 0.983$), while contamination remained absent below 300 larvae/sec (Fig. 5a). We propose that an operational speed for sex sorting would be around 60 larvae/ sec (which corresponds to 200 larvae/mL in the reservoir water with our instrument settings) with a mean recovery rate of $69.6 \pm 1.1\%$ and no female contamination detected among more than 6500 sorted males. At this speed, a small COPAS reservoir (250 mL) holds 50,000 larvae that are sorted in <14 min, yielding approximately $17,400 \pm 270$ males (considering a 50:50 sex ratio). Consequently, 100,000 males could be sorted in less than 1.5 h. The same test run was repeated with *Ae. albopictus* larvae and gave comparable results (Fig. 5a).

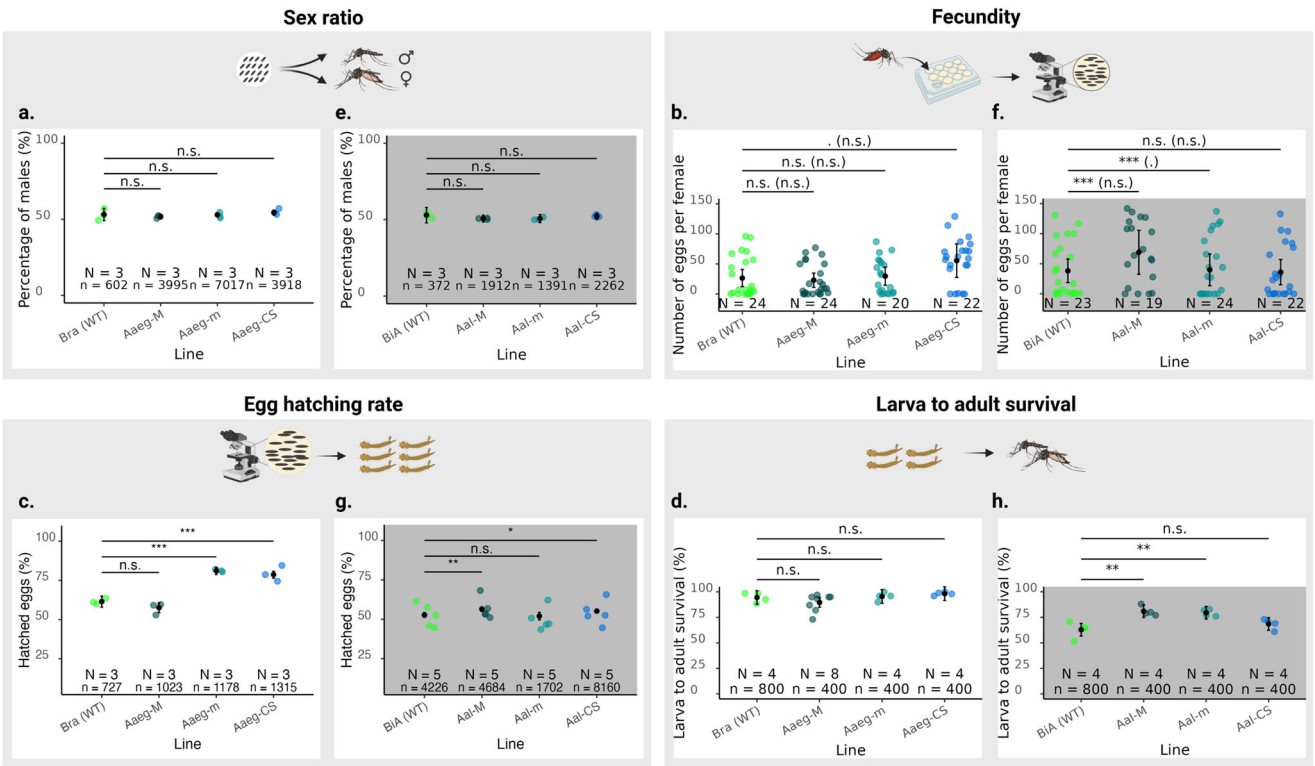

**Fig. 3 Fitness of the genetic sexing strains and of the intermediary colony yielding non-transgenic males.** In all panels, the black dots with vertical bars represent the mean value and the 95% CI in the *Ae. aegypti* and *Ae. albopictus* assays, respectively. The large coloured dots around estimates represent the mean values of each replicate (*N*). Significant differences between the lines are specified in the figure (n.s.: no significant difference. *p-value < 0.1, **p-value < 0.01, ***p-value < 0.001). The percentages of males were compared using a generalised linear model with a binomial distribution in the *Ae. aegypti* (**a**) and *Ae. albopictus* (**e**) lines. Comparisons of the mean number of eggs laid by an individual female were analysed using a hurdle model with a negative binomial distribution in the *Ae. aegypti* (**b**) and *Ae. albopictus* (**f**) lines. Significance of the hurdle model is displayed in two parts: probability of the non-zero values and probability of attaining value 0 (between parentheses). Comparisons of the percentage of egg hatch were analysed using a generalised linear model with a binomial distribution in the *Ae. aegypti* (**c**) and *Ae. albopictus* (**g**) lines. Comparison of survival from first instar larvae to adult stages were analysed using a linear model with the assumption of residual normality in the *Ae. aegypti* (**d**) and *Ae. albopictus* (**h**) lines. Figure designed on BioRender.com.

**Assessing the cost-efficiency of using transgenic GSS as compared to standard sex sorting in SIT programs.** By allowing the separation of males from females at the first larval stage (L1), the numbers of subsequent larval stages and pupae to be reared in a mass-rearing facility is greatly reduced. However, given the recombination rates of our transgenic lines, extra colonies that would be double-checked to remove all contaminants (referred to as 'filter colonies') are required in the mass-rearing facility. Additionally, expensive fluorescence-sorting devices such as COPAS have to be acquired. Adapting the 'FAO/IAEA INTERACTIVE SPREADSHEET FOR DESIGNING MOSQUITO MASS REARING AND MALE HANDLING FACILITIES'[34] (see Methods and Supplementary Data 3), we estimated the number of insects to be reared, the facility sizes, the workforce, the construction, equipment and consumable costs *etc*. These estimates were compared between standard rearing and sorting procedures (hereafter referred to as 'Size manual' for manual sorting of pupae based on their size dimorphism[13], and 'Size auto' for automated sorting of pupae developed for the IIT-SIT in Guangzhou, China[11]) as well as the use of a single GSS (sorting transgenic males, referred to as 'GSS') and the use of two GSS combined for sorting non-transgenic males (referred to as 'GSS-CS', 'CS' standing for 'Crossing Scheme'). We considered the case of GSS-CS (to produce non-transgenic males) separately from that of GSS as it requires two rearing colonies (one for each strain to be used) and two filter colonies instead of one (see the 'Mass rearing processes' schemes in Supplementary Data 3),

which translate into more mosquitoes to be reared and requiring extra space and equipment. For all methods, different release scales (namely 5, 10, 20, 50 and 100 million males to be released per week) were tested according to what is currently being done in genetic control field trials against *Aedes* (5 and 10 M/week) and what would be operationally relevant for a broader *Aedes* control[35]. We observed that both GSS and GSS-CS allow important reductions in the number of insects to be reared in comparison to the standard, no matter the release scale and including the filter colonies (GSS *vs.* Size manual = −65% larvae and −25–30% adults, GSS-CS *vs.* Size manual = −46% larvae and −8–17% adults, see Fig. 5b and Supplementary Data 3). These reductions translate into decreases in the mass-rearing facility size and therefore in the construction cost: facility size and construction cost are estimated to be reduced by 22–43% in GSS and 12-28% in GSS-CS compared to Size manual (Fig. 5c). Given the currently high costs of COPAS devices (from recent quotes, we considered $40k for a refurbished device), equipment costs are increased in GSS-CS compared to the manual sorting of pupae (+9 to +18% yearly, Fig. 5d) while GSS alone is estimated to have the lowest equipment cost from 10 M males/week (−5 to −16% yearly in comparison to Size manual, Fig. 5d). However, the consumable costs, including insect diet costs, are also reduced and allow important savings starting at 20 M males/week and above (GSS *vs.* Size manual = −54%, GSS-CS *vs.* Size manual = −37%, with >$50,000 savings at 20 M males/week and above, Fig. 5e). In total, considering construction, equipment, diet and

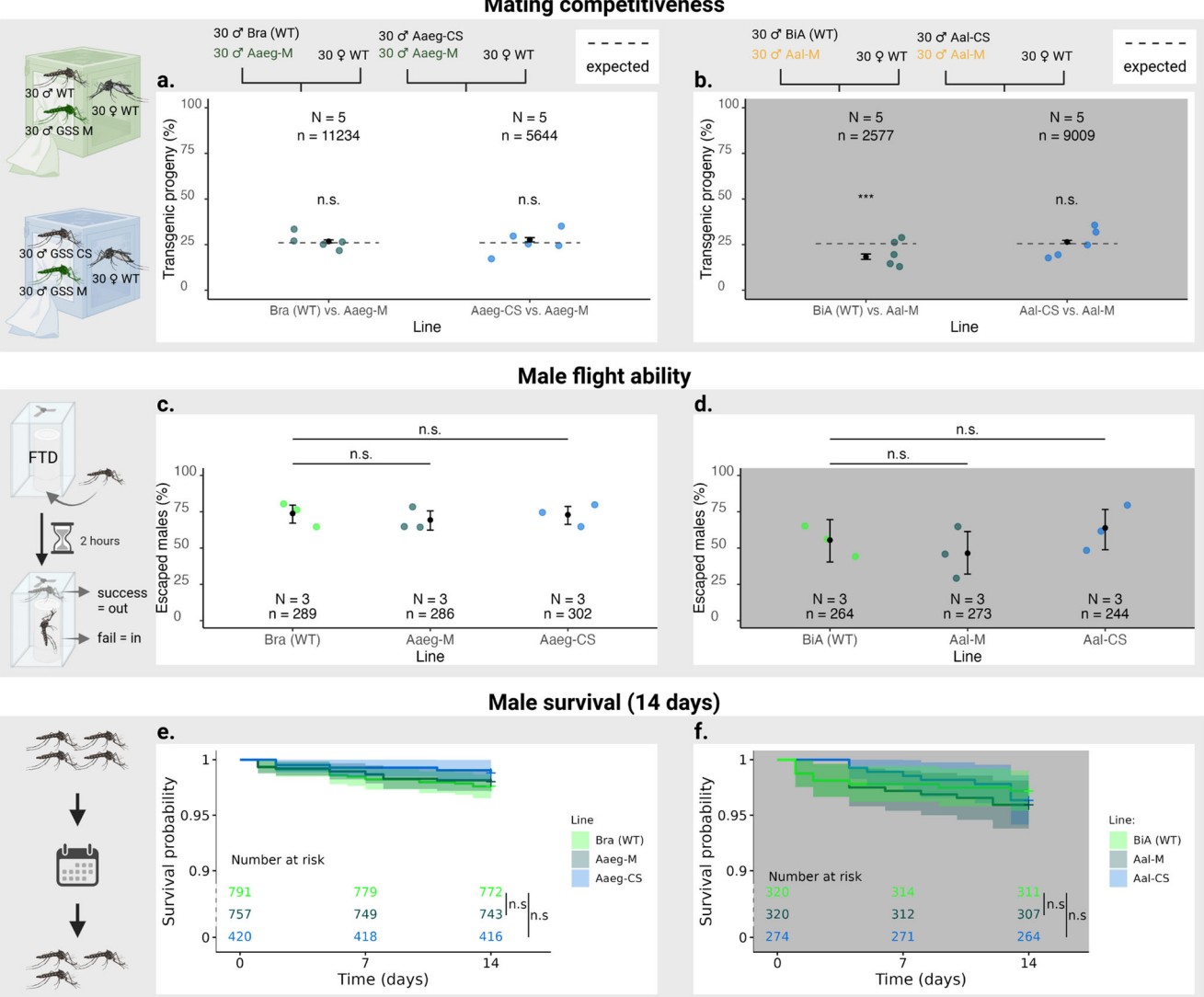

**Fig. 4 Fitness of transgenic and non-transgenic males compared to wild type.** Black dots with vertical bars represent the mean value and 95% CI in the competitiveness and male flight ability assays. The thin coloured dots in the squares above and below the estimates represent individual data points (n), while the large coloured dots around the estimates represent the mean values of each replicate (N). Significant differences between the lines are indicated in the figure (n.s.: no significant difference. *p-value < 0.1, **p-value < 0.01, ***p-value < 0.001). The competitiveness of males of the Aaeg-M line relatively to Bra (WT) or Aaeg-CS in *Ae. aegypti* (**a**) and the competitiveness of males of the Aal-M line relatively to BiA (WT) and Aal-CS in *Ae. albopictus* (**b**) were compared by measuring the number of transgenic and non-transgenic L1 larvae in the progeny of each of the N = 5 replicates. Results were compared to the percentage expected if transgenic males were as competitive as non-transgenic males (dashed line) and were analysed using a generalised mixed-effects model with a binomial distribution and replicates as a random effect. The flight ability of transgenic and non-transgenic males compared to the wild-type in *Ae. aegypti* (**c**), and *Ae. albopictus* (**d**) was tested by N = 3 replicates in a reference flight test device ("FTD"). The results were analysed using a generalised mixed-effect model with a binomial distribution and replicates as a random effect. Survival of adult males 14 days after emergence was monitored in *Ae. aegypti* (**e**) and *Ae. albopictus* (**f**) lines. N = 4 replicates were performed for each line. All values being above 90%, the y-axis is discontinuous between 0 and 90% for better visualisation. Male survival at 7 and 14 days was compared using the log-rank test and neither comparison was significantly different. Figure designed on BioRender.com.

consumable costs, GSS is predicted to be the most economical option at all tested throughputs, while GSS-CS starts to be more economical compared to manual sorting of pupae from a 10 M males/week throughput (Fig. 5f). Moreover, the workload, the cost of which can only be estimated on a country-by-country basis, is reduced by 29–38% with GSS and 18–27% with GSS-CS (Fig. 5g). For all comparisons, including equipment cost, the most expensive option is the automated sorting of pupae, as the devices are expensive (e.g. $40k for the automated sorter from[11], J. Bouyer) and the number of insects to be reared is high (Fig. 5b–g).

## Discussion
Here we present the first genetic sexing strains (GSSs) for *Ae. aegypti* and *Ae. albopictus* based on m/M-linked transgenic fluorescent markers. We demonstrate that the M-linked Aaeg-M and Aal-M strains can be used alone to sex transgenic first instar larvae and that the M- and m-linked strains of each species can also be combined to produce a non-transgenic male population by two rounds of sex-sorting (of the parental strains and of the cross' offspring). In *Ae. aegypti*, transgenic males from the M-linked strain and non-transgenic males from the crossing

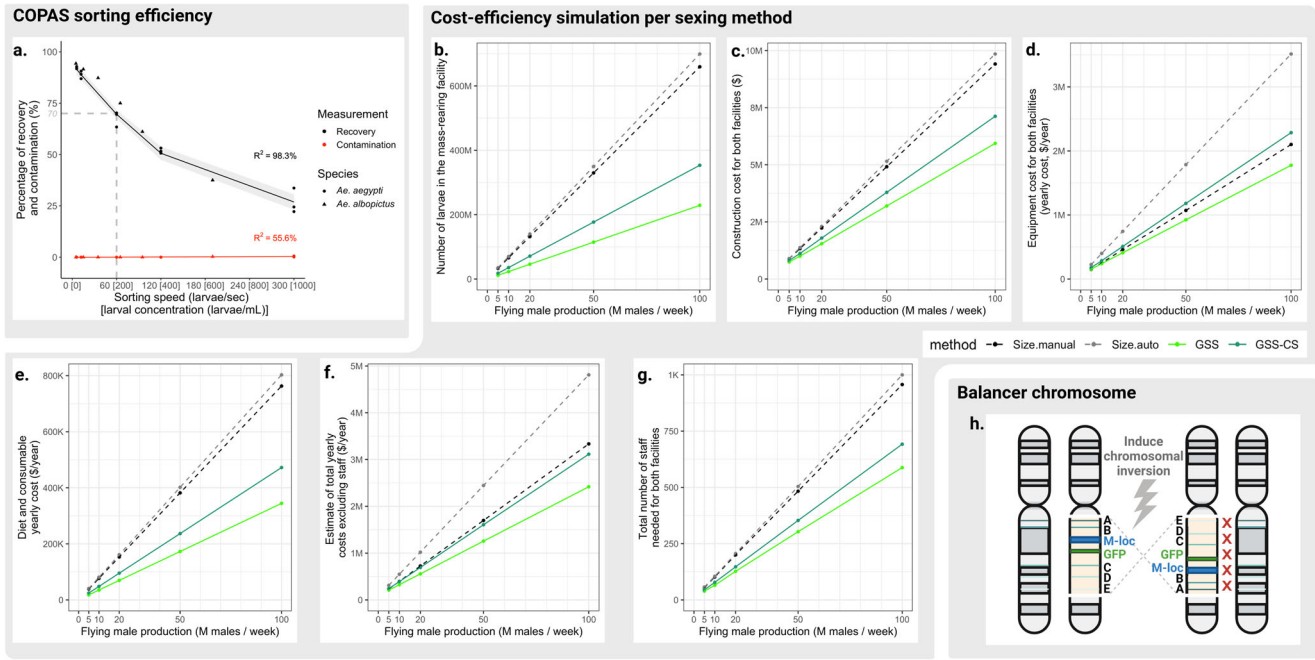

**Fig. 5 Evaluation of the feasibility of using GSS in *Aedes* SIT, and suggested improvement. a** Evolution of male recovery and female contamination rates as a function of COPAS sorting speed. 10,000 *Ae. aegypti* L1 larvae from the crossing scheme were COPAS-sorted at different concentrations, resulting in different sorting speeds. Percentage of recovery (black dots) and percentage of contamination (red dots) were measured on three replicates for each sorting speed. Recovery values were fitted using a degree-two polynomial regression ($R^2 = 0.983$) and contamination values were fitted using a linear regression ($R^2 = 0.556$). The model predictions were plotted by black and red lines, respectively, with 95% confidence intervals ribbons. The same experiment was repeated with 4000 *Ae. albopictus* L1 larvae from the crossing scheme and is shown on the same graph. In this species, percentage of recovery (black triangles) and percentage of contamination (red triangles) were measured on a single replicate for each sorting speed. **b–g** Estimate of the cost-efficiency of Genetic Sexing Strains for *Aedes* SIT by simulating the construction of a mass rearing and a release facility with different sexing systems. Included sexing systems were the manual sorting of pupae using glass plates, "Size manual" (dashed black lines), the automated sorting of pupae with robotic glass plates, "Size auto" (dashed grey lines), the automated sorting of L1 larvae from a GSS using COPAS, "GSS" (light green lines), and the combination of two GSS for automated sorting of non-transgenic L1 larvae, "GSS-CS" (dark green lines). Different production objectives were considered: release of 5, 10, 20, 50 and 100 million males per week for 52 weeks per year. **b** Comparison of the number of larvae present at any time in the rearing facility. **c** Comparison of the construction cost in US dollars for both the mass rearing and the release facilities. **d** Comparison of the equipment cost in US dollars for both the mass rearing and the release facilities (yearly cost estimated by dividing the cost of each equipment by its lifespan). **e** Comparison of the diet and consumable cost per year in US dollars for both the mass rearing and the release facilities. **f** Sum of the yearly costs for construction (estimated lifespan = 20 years), equipment, diet and consumables in US dollar. **g** Comparison of the number of staff (labourers, managers etc.) needed for both the mass rearing and the release facilities. **h** Proposed chromosomal inversion that could prevent recombination between the fluorescence marker and the sex-locus. Figure designed on BioRender.com.

scheme were found to be at least as fit and competitive as wild-type males of the same laboratory strain. In *Ae. albopictus*, a mild competitiveness reduction was observed in Aal-M transgenic males as well as in Aal-CS non-transgenic males. The reason for this is unclear, especially since Aal-CS males are non-transgenic. At a speed of 60 larvae/second using a COPAS device, we estimate that 100,000 males can be obtained in under 1.5 h with the current settings, recovering 70% males and with a female contamination of 0.01–0.1%.

Unexpectedly, we observed rare recombination events between the transgene and the M-locus in the Aaeg-M line after screening tens of thousands of larvae. In this line, the transgene is located approximately 9 Mbp from *Nix*, which remains well within the presumed non-recombining region[30]. Compared to the probability of recombination between two markers separated by the same distance on ordinary loci (3.4–4.7% according to the latest genomic information[27]), these results confirm that recombination is strongly reduced in this region, but not totally suppressed. Although not yet observed, the Aaeg-m line is expected to recombine at the same frequency as Aaeg-M, as both transgenes are inserted in the same target gene at a similar distance from their respective sex loci. Unfortunately, the recombination events

imply that these GSSs will require regular purification of the parental colonies to avoid a gradual accumulation of recombinant individuals. Of note, such recombination events are easy to spot in single-sex batches of pupae due to size dimorphism and protandry, which will facilitate regular quality control of each strain.

Although our initial attempts at knocking-in a GFP marker into the large intron of *Nix* failed, improved GSSs may be constructed by targeting regions closer to or within the sex loci, especially in *Ae. aegypti* which benefits from a fully assembled genome. In *Ae. albopictus*, rare recombination events were also found in Aal-M amongst more than 10,000 individuals screened over 10 generations. For more accurate genome editing, better assembly of the first chromosome would be necessary in this species. The possibility of genetic instability under mass-rearing conditions and the appearance of aberrant sex recombinants have already been reported in the development of GSSs in other insect model organisms[36]. Alternatively, the creation of viable chromosomal inversions spanning the sex-determining locus and the fluorescent transgene, akin to a balancer chromosome, could help suppress crossing-over (Fig. 5h)[37,38]. However, inversions will need to be carefully analysed to ensure that they don't introduce unwanted fitness costs into the sexing strain which has been

observed in other species[39]. Finally, we have recently reported that transgenic *Ae. albopictus* pseudo-males masculinised by ectopic expression of *Nix* can be used as the Aal-M GSS in which the fluorescent marker will show an absolute linkage to sex[26]. This type of GSS is particularly interesting for the production of non-transgenic males, since in this case the wild-type females derived from the Aal-M line and used in the crossing scheme are both non-transgenic and devoid of any recombinant contamination.

The female contamination rates of our GSSs (0.01 to 0.1% depending on recombination events) are higher than with the Verily's multi-step sorter[9], but much lower than with manual[13] or automated[11] glass sorters. Of note, up to 0.3% of contaminant females were considered acceptable in recent SIT-IIT assays[11,40] while up to 1% is tolerated for SIT alone[5,35,41–43]. Moreover, contaminant females would be subjected to the same irradiation dose as males, which has been proven to cause 100% sterility in females[44]. Therefore, the main concern regarding these contamination rates is that contaminant females have to be removed from the colonies in the mass-rearing facilities so that they do not accumulate. This has been considered in our simulations by including filter colonies that would be sex-sorted as larvae and double-checked as pupae. In cases where closer to 0% contamination is required in the release batches, a quality control step using automated pupal sorters could also be implemented prior to adult emergence.

Compared to current sex-separation methods in *Aedes* mosquitoes[9,11,13,45,46], these GSSs allow a higher male recovery than any sorting method based on pupal size (about 70% *vs.* 30%). Our system offers a sorting speed similar to that of the automated glass sorter, which is higher than that of manual size sorters and multi-step sorters. Compared to all of these currently available methods, the main advantage of our GSSs is to allow sorting at the earliest stage of development, which results in significant savings in space, time and cost otherwise spent on rearing unnecessary individuals. Moreover, reducing the number of females reaching the pupal stage in the mass-rearing facility decreases the biting nuisance for the workers, a problem frequently observed with pupal sorting methods, due to early adult emergences.

Additionally, we propose a crossing strategy that proved to be effective in obtaining non-transgenic males. This strategy could be particularly useful in countries where genetically modified insects are not strictly banned but where releasing transgenic mosquitoes might cause strong public concern. Compared to sorting pupae by size using glass plates or metal sieves, it has the advantage of automating all sorting steps and achieving much lower female contamination rates. Although it requires the rearing of two strains plus the sorting of males in the mass-rearing facility, this still represents a reduction of approximately 46% in the number of mosquito larvae raised for current size-based manual sortings. Despite the high cost of the sorting devices and the higher insect numbers than for GSS, this method is predicted to be more cost-effective than the automated sorting of pupae at all throughputs, and than manual sorting of pupae when exceeding a production throughput of 10 M males/week.

The release of non-transgenic males from crossbreeding schemes may in fact be more appropriate than the release of transgenic males in many countries, since GMO mosquito releases face more public opposition and regulatory constraints[47]. Therefore, costs related to public information and consultation campaigns, stakeholder engagement and compliance with regulatory processes may be reduced. These aspects need to be considered as in the past, lack of information and consultation has led to such strong public opposition that some release programmes have had to be cancelled[47]. Of note, recombination in

our GSSs would result in contamination of released males by non-transgenic irradiated females at a rate equal to the recombination rate; however, this release of sterile females would still be much lower compared to current methods.

COPAS sorting errors during the purification of non-transgenic males were extremely rare in our pilot experiments. However, such contaminations may occur due to occasional technical problems, and would result in the release of a small percentage of transgenic mosquitoes amidst the non-transgenic. This should be absolutely avoided in settings where GMOs are banned. An additional quality control step to eliminate any residual GFP fluorescent larvae from the sorted larvae could be achieved by a second COPAS run. It could also be achieved by visual verification of the sorted larvae collectively under a fluorescence microscope, as GFP positives are easy to spot and eliminate, even among many wild-type larvae. Of note, GSSs with an m-linked fluorescent marker show less obvious sex separation than M-linked GSSs. However, contamination of the sexing cross with females homozygous for the transgene yields transgenic progeny that is subsequently eliminated when sorting the non-transgenic males. Also, the irradiation of all mosquitoes before release will prevent leaking of the transgenes into the target population whatsoever.

Although this work has focussed on the production of *Aedes* males for SIT, the same procedures for the production of transgenic or non-transgenic male could be directly applied in other mosquito control approaches such as IIT, RIDL, pgSIT or other genetic control methods currently under development.

In conclusion, the four GSSs developed in this work represent very promising tools for improving the cost-efficiency of *Aedes* mass-rearing in genetic control programmes. According to our laboratory pilot tests, a facility harbouring two COPAS machines could produce 1,000,000 male larvae per day in a 7.5-hours regime. Considering that no females are reared, space constraints would be eased. Moreover, this technique would also reduce the production costs associated with the need to reach low or no female contamination, facilitating compliance with local regulations. Our results suggest that COPAS sorting of our GSSs would allow a rapid, cost-effective, and safe mass-production of male mosquitoes for SIT and other genetic control methods, with flexible up-scaling possibilities according to local needs, resources and regulations.

## Methods

**Mosquito rearing**. The *Ae. albopictus* wild-type colony (called BiA) was established from larvae collected in Bischheim (France, 48.36°N 07.45°E) in 2018. *Ae. aegypti* from the Bangkok genetic background obtained from MR4-BEI were used for molecular cloning and genetic engineering. *Ae. aegypti* strain BgR9, derived from Bangkok, and expressing *Cas9* under the control of the Exuperentia promoter and a *DsRed* reporter gene under the *Ae*PUb promoter was used for CRISPR/Cas9 knock-in experiments (plasmid sequence provided in Supplementary Data 4). The Bra *Ae. aegypti* strain originated from Juazeiro, Brazil and was provided to the IPCL in 2012 by Biofabrica Moscamed, a collaborative centre of the IAEA on the development of the SIT against mosquitoes. It was used for backcrossing the sexing transgenes.

Mosquitoes were maintained in standard insectary conditions (25–28 °C, 75–80% relative humidity, 14-h/10-h light/dark cycle). Larvae were reared in 35 cm square pans in 1 L distilled water and daily fed ground TetraMin fish food. Adult mosquitoes were kept in 16 × 16 × 16-cm cages and provided a 10% sugar solution. Females were blood-fed on anesthetised mice. Eggs were laid on humid kraft paper and hatched or dried after 3 days.

**Plasmid construction**. Plasmid construction employed standard molecular biology procedures including PCR amplification of genomic DNA using Phusion™ High-Fidelity DNA Polymerase (F530, Thermo Scientific, France), cloning with NEBuilder® HiFi DNA Assembly (New England Biolabs, France), GoldenGate cloning in destination backbones using Anza 36 Eco31I restriction enzyme (Thermo Fisher Scientific, France), transformation in competent *Escherichia coli* bacteria, and Miniprep or Endo-Free Midiprep plasmid purifications (Macherey Nagel, France). For CRISPR/Cas9 knock-in in the *Ae. aegypti* M and m loci, we used plasmid pX4

and pX3 as repair donors, respectively (Supplementary Data 4, Addgene #183903 and #183904). For piggyBac integration near the Ae. albopictus M locus yielding the Aal-M strain, we used plasmid ppBAalbNixE1E3E4 PUb-YFP (Addgene #173666 described in[26]). For random piggyBac insertion near the m locus, we used plasmid ppBExu-Cas9-sv40, the lox cassette of which (encompassing Cas9 and PUb-DsRed) was subsequently excised from the genome by injection of a plasmid expressing Cre recombinase under the control of the PUb promoter, and replaced by inserting plasmid pENTR-attB-PUb-GFP (Addgene #183911) into the attP site by co-injecting it with a plasmid expressing integrase from phage PhiC31 (Addgene #183966).

**Microinjection**. Injection mixes were composed of 400 ng/µL of DNA in 0.5x PBS. piggyBac injection mixes were prepared as described in[20]. For CRISPR-Cas knock-ins, the initial mix injected into the BgR9 Ae. aegypti Cas9-expressing line comprised 84 µM gRNAs, 100 ng/µL repair plasmid and 2 µM Scr7. Given the very low number of transgenics obtained (1 individual out of >20,000 screened larvae), we later injected the repair donor plasmid (190 ng/µL) with 5 µM Scr7 and three plasmids expressing different gRNAs under the control of different AeU6 promoters (70 ng/µL each plasmid). This method gave significantly higher knock-in rates (25 PUb positive larvae out of about 4000 screened).

Embryo microinjection was performed essentially as published using a Nikon Eclipse TE2000-S inverted microscope, an Eppendorf Femtojet injector and a TransferMan NK2 micromanipulator[18] with the following modifications: eggs were hatched 3–8 days post-injection and first instar larvae were screened under a Nikon SMZ18 fluorescence microscope, only G0 larvae showing transient expression of the reporter gene were retained for subsequent outcross.

**Ae. aegypti transgenesis**. For targeting the sex loci, we exploited the work of Fontaine et al. showing a cluster of markers inside a non-recombining 63 Mbp region encompassing the sex loci with high male-female genetic differentiation and male heterozygosity consistent with Y chromosome-like null alleles[30] (Supplementary Table 2). DNA sequences of these markers with putative Y-linked-like null alleles were blasted on the assembled AegL5 genome[27] to identify a 4.9 Mb region extending from 160,092,339 bp to 165,014,624 bp on chromosome 1 in the AegL5 assembly. Within this non-recombining region, we picked the gene AAEL019619, located ~9Mbp away from Nix. We used the CRISPR/Cas9 system to knock-in a 2.3 kb fluorescence marker cassette (eGFP marker under the Ae. aegypti poly-Ubiquitin promoter - PUb) flanked by 850-960 bp homology arms either within the fifth, 9 kb-intron of this gene (pX3 construct), or to create a synthetic intron within exon 5 (pX4 construct). We obtained a first M-linked strain for Ae. aegypti following the injection of 200 eggs of the BgR9 strain, which expresses Cas9 under control of the Exuperentia promoter, with the pX4 repair plasmid and two synthetic gRNAs (GATGAATCATGGGGCGCCT[GGG] and GATTCATCAATCA ACGGAG[CGG], brackets indicate the Protospacer Adjacent Motif, PAM). About 30 G0 larvae showing transient GFP expression were crossed en masse to an excess of wild-type mosquitoes. Out of >20,000 G1 larvae screened, a single transgenic male larva was found, raised to adulthood, and crossed to WT females. Its progeny was composed of 100% eGFP positive sons and 100% eGFP negative daughters, indicative of M linkage. We termed this Ae. aegypti M-linked strain Aaeg-M. Upon injection of 600 BgR9 eggs with the pX3 repair plasmid and a mixture of three plasmids expressing gRNAs (GTGGCATAGCGCCGTGTGGA[GGG], GTCTT AAATGAAAGAGGCG[AGG], GTATCATGCGTATTGCGAG[AGG], brackets indicate the PAM) under the control of three different U6 promoters (gRNA cloning vectors: Supplementary Data 4), 43 larvae with transient eGFP expression were recovered in G0. Resulting 20 males and 20 females were crossed en masse to adults of the opposite sex. 25 transgenic G1 larvae were obtained, 10 of which were males carrying an M-linked insertion, 4 were males carrying an m-linked insertion and 11 were females carrying an m-linked insertion. Proper insertion site in AAEL019619 was confirmed by PCR in all tested individuals, which revealed that in some of these mosquitoes the whole repair plasmid had integrated. From a single male devoid of the plasmid backbone, an Ae. aegypti m-linked strain named Aaeg-m was established and further characterized. Both Aaeg-M and Aaeg-m lines were backcrossed into a Brazilian (Bra) genetic background for 7 generations.

**Ae. albopictus transgenesis: obtaining M-locus linked strains**. Following several unsuccessful attempts to knock-in a fluorescence marker near the Ae. albopictus Nix gene using the CRISPR/Cas9 system, we took advantage of the frequent spontaneous insertion near the M locus of piggyBac transposon constructs carrying Nix DNA sequences that we previously reported[26]. In the course of several transgenesis experiments, using piggyBac constructs carrying Nix sequences, we obtained 12 independent transgenic insertions with obvious M-linkage of the transgenesis reporter (100% fluorescent males) in addition to several Nix-expressing masculinized female lines and to non-M linked, non-masculinizing insertions. From the third generation on, these lines were amplified to confirm their M-linkage and estimate their recombination rate. Six lines showed no recombination out of >700 individuals screened in each line until the sixth generation (Supplementary Table 3). Most other M-linked lines also showed 100% fluorescent males but contained additional, non-M-linked multiple insertions and were discarded. Based on line fitness and COPAS profiles, we selected a YFP line termed Aal-M for

further work as an Ae. albopictus M-linked strain. Additional M-linked lines marked with OpIE2-GFP or Pub-DsRed showing no recombination to date, carrying a docking attP site for subsequent transgenesis, and representing additional GSSs are also being maintained to date but were not further characterized in detail.

**Ae. albopictus transgenesis: obtaining an m-locus linked strain**. We first attempted to obtain an m-linked fluorescence marker by screening through >120 random piggyBac insertions. For this, we screened our existing collection of Ae. albopictus transgenic lines for m/M-linkage and constructed an additional library of piggyBac constructs expressing various fluorescence proteins (eGFP, mTurquoise2, YFP or DsRed) under the control of promoters showing distinct expression patterns (3xP3, OpIE2, Ae. aegypti PUb, Drosophila melanogaster actin5C). By performing multiplex injections of these piggyBac plasmids together, we obtained lines carrying up to six distinct insertions as indicated by their colours and patterns of fluorescence. Out of 40 independent founder transgenic individuals screened (most of which carrying multiple insertions), only two insertions appeared m-linked: one carried an OpIE2-mTurquoise2 marker gene with a recombination rate of about 3%, and the other, called m-albR9, carried a Cas9 transgene with a DsRed reporter gene and a recombination rate of 0.1%. We exploited the latter, in which transgenes are flanked by two lox sites adjacent to an attP docking site. We injected m-albR9 eggs with a plasmid encoding Cre recombinase to excise Cas9 and DsRed transgenes, as Cas9 is undesirable in a GSS. From successfully excised individuals, we established an m-linked attP docking line, mX1, carrying no further transgene. In a second step, an attB-containing plasmid carrying a PUb-eGFP marker cassette was integrated into the attP site. The m-linkage of this new strain was verified by crossing eGFP males to WT females and screening their offspring. The new Ae. albopictus m-linked line, named Aal-m was made hemi/homozygous and amplified.

**Automated larva sorting**. Automated sorting of L1 larvae depending on their fluorescence level was performed using a large-object flow cytometer named Complex Object Parametric Analyzer and Sorter (COPAS SELECT; Union Biometrica, Belgium) as published[16], with the provided Biosort software. Sorting accuracy was controlled under a Nikon SMZ18 fluorescence microscope.

**Analysis of COPAS outputs**. To determine the exact number of larvae in COPAS clusters, we built a clustering algorithm based on manual outlier screening of particle size and fluorescence distribution, and an automated clustering algorithm. All data handling and statistical calculations were performed using the R statistical software version 4.1.0[48]. All R scripts used for this project can be found on Zenodo[49]. Briefly, COPAS raw data were filtered by the user via the visual examination of the individuals' size measurements (i.e., "log(EXT)" and "log(-TOF)") to remove outliers (e.g., egg debris and dust). A second filtering was performed on the two first axes of a Primary Component Analyses (PCA) applied to the size variables (EXT and TOF). A third manual filter was applied by examination of the individuals' fluorescence (i.e., "log(first fluorescence)" and "log(second fluorescence)"). A final filtering was performed on the two first axes of a PCA applied to fluorescence variables. PCA analyses were performed using the R function 'prcomp' from the package 'stat'[48] and the PCA outputs were extracted using the R package 'factoextra'[50]. Filtered data were clustered using the R function 'kmeans' from the R package 'stat'[48]. Data were plotted using the R packages 'ggplot2' and 'ggpubr'[51,52].

**Targeted sequencing method**. Genomic DNA was extracted using NEB Monarch HMW DNA extraction kit (T3060, New England Biolabs, France) according to manufacturer instructions with the following modifications: after addition of isopropanol, precipitated DNA was caught directly on a pipette tip and transferred to a new tube containing 550 µl wash buffer and then to a final tube containing 70% ethanol. DNA was stored in 70% ethanol at 4 °C until use. On the day of use, ethanol was removed and DNA was resuspended in water at 37°C with 500 rpm agitation and gentle pipetting. DNA concentration was then measured using a Nanodrop One device (Ozyme, France).

Sequencing method was adapted from the Nanopore Cas9 Targeted Sequencing (nCATS)[53]. gRNAs were selected in the known transgene region, with their PAMs towards the unknown genomic sequence to be determined. This orientation ensured preferential ligation of Nanopore adapters on DNA strands of interest.

Adapter ligation and library preparation were performed using the LSK109 kit (Oxford Nanopore Technologies, UK) according to manufacturer instructions. Library was loaded on Flongle flow cells. Sequencing, basecalling and alignment of reads to the transgenesis plasmid were performed using the provided MinKNOW v. 21.06.10 software (Oxford Nanopore Technologies, UK). Reads that showed homology to the transposon ends were then aligned to the transposon sequence using ClustalW on Geneious software v9.1.8[54] and one or several consensus sequences were determined. Finally, the consensus sequences were compared to genomic data using the online NCBI BLAST service[55].

This method allowed us to determine the insertion site of the Aal-m transgene. We obtained only two low-quality reads extending by about 20 kb into the genomic sequence adjacent to the transgene, from which we designed a series of PCR primers. Sanger sequencing of PCR products spanning the transposon/genomic junction allowed us to refine 1647 bp of the flanking sequence, provided in

Supplementary Data 1. Analysis of this sequence using NCBI BLAST showed only partial alignment to sequences of the AalbF2 genome[32], presumably corresponding to a repetitive element, suggesting that the m-linked region where the transgene inserted is not represented in the available genome assemblies.

**Assessment of lines' fitness**. In transgenic lines, the sex ratio was measured by COPAS counting of the larvae of each fluorescence. In non-transgenic lines (BiA and Bra), we reared samples to the pupal stage and counted males and females under a binocular microscope based on genitalia examination. $N = 3$ biological replicates were made for all lines.

For fecundity, males and females were allowed to mate for several days. Cages were then offered a blood meal and engorged females were selected. After 2 days, engorged females were shortly anesthetised using $CO_2$ and placed individually in 24-well plates coated with wet filter paper for 2 days. Females were then released and each well was photographed under a Zeiss SteREO binocular microscope. Eggs laid by each female were counted using the ImageJ software[56].

Egg hatching rate was measured on egg samples collected from $N = 3$–5 cages of each line, photographed under a Zeiss SteREO binocular microscope and counted using the ImageJ software[56]. Eggs were immersed in water, placed into a vacuum chamber for 20 to 30 min and allowed to hatch overnight. The following morning, larvae were counted by COPAS.

Larva to adult survival was measured by sorting $N = 4$–8 samples of 100–200 L1 larvae by COPAS and rearing them to adulthood. The number of adults emerging from each batch was counted manually.

**Assessment of males' fitness**. Males' competitiveness was assessed by placing 30 transgenic males with 30 non-transgenic males in a cage with 30 females and measuring the proportion of transgenic progeny in their offspring. If both types of males are of similar fitness, it is expected that the progeny would be composed of ~25% transgenic individuals (transgenic fathers being homozygous), corrected by the line sex ratio. Relative competitiveness was estimated by comparing the observed proportion of transgenic progeny to the theoretical value in $N = 4$ or $N = 5$ replicates.

Males' flight ability was assessed using a flight test device (FTD) as described[33]. Tests were performed on pools of about 100 males with $N = 3$ replicates for each condition. Pools of males were placed in a tight cup at the bottom of several vertical tubes topped by a fan. In order to escape the tight cup, they have to flight up the vertical tubes against the airflow caused by the fan. After two hours, the number of males that escaped versus the number that did not were counted and escape rates were compared between strains.

Males' 2-week survival was measured by counting the daily number of deaths in $N = 4$–8 cages of ~100 males of each line for 14 days.

**Cost-efficiency analysis**. The cost efficiency analyses were performed by comparing the outputs of the 'FAO/IAEA INTERACTIVE SPREADSHEET FOR DESIGNING MOSQUITO MASS REARING AND MALE HANDLING FACILITIES'[34] for different weekly production targets with that of its adapted versions taking into account the use of a GSS or two GSS combined for sorting non-transgenic males (see scheme in Supplementary Data 3). Briefly, these new spreadsheets were designed in a similar way to the original one, but incorporate the GSS-specific structure previously developed in fruit flies that includes separate filter and rearing colonies[36]. In the filter colonies, first instar larvae are sorted using COPAS, and sorted once more at the pupal stage using a classical pupal sorter. These colonies produce the eggs feeding the rearing colonies at each generation. In a single GSS scenario, the rearing colony is sorted as L1s for producing the males to be released. In a GSS-CS scenario, two separate filter colonies produce the eggs for their respective rearing colony, which in turn are sorted and mated to produce the eggs that will be hatched and sex-sorted as L1s for male-only releases. In these scenarios, filter colonies are used at each generation, which is more careful than what is being done in fruit flies despite a higher recombination rate[36]. Input parameters can be read in Supplementary Data 3. Of note, the costs mentioned in the spreadsheet have been implemented in euros (€) and updated in July 2022. These costs are average observations from European SIT assays and were revised by IAEA researchers. To facilitate downstream analyses, these Excel spreadsheets were translated into an R[48] function. This function uses common R-base[48] functions as well as functions from the 'purrr' and 'tidyverse' packages[57]. It was permanently deposited on Zenodo[49]. The output costs were converted in US dollars ($).

**Statistics and reproducibility**. All data statistical analyses were performed using the R statistical software version 4.1.0[48], the R scripts used for this study can be found on Zenodo[49].

The difference of sex ratio and egg hatching rate between the lines of each species was modelled using the generalised model with binomial distribution and residuals distribution normality assumption (R function 'glm'[48]), and pairwise comparison of the lines was tested using a Tukey HSD test (R function 'glht'[58]). The difference in fecundity (number of eggs laid per female) between the lines of each species was modelled using a hurdle model with a negative binomial distribution. Significance of the hurdle model is obtained in two parts: the first part being the probability of attaining value 0 and the second part being the probability of the non-zero values. The differences in larval survival between the lines of each

species was modelled using linear models with residuals distribution normality assumption (R function 'lm'[48]), and tested using a Tukey HSD test (R function 'TukeyHSD'[48]). The difference in male competitiveness and flight ability between the lines of each species was modelled using the generalised mixed-effect mod with binomial distribution and residuals distribution normality assumption (R function 'glmer' from the package 'lme4'[59]), where technical replicates were set as random factor. Pairwise comparisons between lines were tested using a Tukey HSD test (R function 'glht'). Male adult survival during the first 14 days was modelled using Cox regressions and Kaplan–Meir formula (function from the R packages 'survminer', 'survival'[60–62]). Differences in survival at 7 and 14 days were modelled using linear models with residuals distribution normality assumption (R function 'lm') and tested using a Tukey HSD test (R function 'TukeyHSD'). Model performances were assessed using the R function 'check_model' from the package 'performance'[63]. The packages used, data handling procedures, model structure, fit and performances can be found in Supplementary Data 2.

**Reporting summary**. Further information on research design is available in the Nature Portfolio Reporting Summary linked to this article.

## Data availability

All plasmid sequences are available in Supplementary Data 4. Plasmids are available from Addgene under reference numbers #173666, #183903, #183904, #183911, #183912, #183913, #183914 and #183966. Mosquito strains are available upon request from EM. The source data to generate the charts in Figs. 3 and 4 are provided in Supplementary Data 2. The source data for Fig. 5 is provided in Supplementary Data 3 ('Initial Parameters' section). Datasets can also be downloaded from Zenodo[49]. Any remaining information can be obtained from the corresponding authors upon reasonable request.

## Code availability

The R function replicating all four spreadsheets, the COPAS analysis code and the code for replicating our statistical analyses are available on Zenodo[49]. All other code used in this study is available from the authors upon reasonable request.

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

## Acknowledgements

The authors thank Dr. H. Maiga and Dr. W. Mamai from the International Atomic Energy Agency (Vienna, Austria) for the revision of the SIT-related costs, Nathalie Schallon and Amandine Gautier for insectary operation, and Dr. Stéphanie Blandin for continuous support and scientific discussion.

This study was funded by EU ERC grant CoG—682387 REVOLINC to J.B. The contents of this publication are the sole responsibility of the authors and do not necessarily reflect the views of the European Commission. Mosquito production and insectarium operation were supported by Agence Nationale de la Recherche grant #ANR-11-EQPX-0022 and contrat triennal "Strasbourg capitale européenne" 2019-2021. Part of the mosquito experiments have been performed on the Baillarguet insectarium platform, member of the National Infrastructure EMERG'IN and of the Vectopole Sud network (http://www.vectopole-sud.fr/). The Baillarguet insectarium platform is led by the joint units Intertryp (IRD, Cirad) and ASTRE (Cirad, INRAE). E.M. received funding from Agence Nationale de la Recherche through grants #ANR-19-CE35-0007 GDaMO and # 18-CE35-0003-02 BAKOUMBA. R.B. was supported by the International Research Training Group TreeDì funded by the Deutsche Forschungsgemeinschaft (DFG, German Research Foundation)—319936945/GRK2324. O.S.A was supported by NIH awards (R01AI151004, 5R21AI156018, 5RO1AI148300).

All figures were designed on BioRender.com under paid license.

## Author contributions

C.L., T.B., J.B. and E.M. designed research. A.F. contributed to genomic data. C.L., M.B., R.P.O. and E.M. performed research. C.L., R.P.O., R.B., A.F., O.S.A., R.A.H. and E.M. analysed data. R.B. and R.A.H. contributed analytic tools. C.L. wrote the paper with inputs from all authors.

## Competing interests

O.S.A. is a co-founder of Agragene, Inc. and Synvect, Inc. with an equity interest. The terms of this arrangement have been reviewed and approved by the University of California, San Diego in accordance with its conflict of interest policies. The remaining authors declare no competing interests.

## Ethics

We support inclusive, diverse, and equitable conduct of research. Animal care was in accordance with CNRS guidelines and approved by the CREMEAS ethics committee.
