## [Peer Review File · Communications Biology]

Reviewers' comments:

Reviewer #1 (Remarks to the Author):

Lutrat et al. describe a set of transgenic *Aedes aegypti* and *Aedes albopictus* transgenic strains that, via linkage of fluorescent markers to either the M or m loci, allow using crossing and fluorescent larval sorting to generate transgenic or non-transgenic single sex mosquito populations. Such populations are the foundation for genetic control strategies based on sterility and the authors assess the limitations and cost-effectiveness of their approach compared to other strategies.

This is an impressive piece of work and clearly the cumulation of many years of mosquito wrangling. I appreciated that the authors tried to make this large collection of strains, data and results accessible via great figures and that they provide data and code that allows the reader to evaluate their findings and statistical models.

Because the paper summarizes a large set of experiments where events were screened for over several generations and in different ways it is sometimes descriptive and glances over some of the steps in describing how certain things were done obtained. I find this acceptable in this case as the huge amount of work summarized here is a tour de force of *Aedes* transgenesis. Nevertheless, I made some suggestions below to make the paper clearer and more accessible.

I could not fully evaluate the economic model presented by the authors as this is beyond my expertise but found that part of the paper among the most interesting.

Overall, the paper can be accepted with minor revisions, it is an impressive piece of work that will find its place in the genetic control literature.

Specific suggestions:

Abstract

“error-prone, and wasteful sex separation requirement”

Requirements are not wasteful or error-prone. Particular methods are.

“Scaling-up would allow the sorting of 100,000 first instar male larvae in under 1.5 hours with an estimated 0.01-0.1% female contamination.”

This sentence doesn't make sense. Scaling-up can always go further. If you refer to particular numbers of insects achievable you need to refer to specific setups (x number of COPAS machines etc) when talking about scaling up. But that is best not done in the abstract.

Results

“targeting the AAEL019619”

Perhaps a short description of this gene and the rationale for targeting it is warranted.

“stimulated by sequences derived from the Nix gene and cloned in the piggyBac transgenesis plasmid”

Perhaps a little more description is warranted here.

“Sequencing revealed that the transposon had landed”

What method was used here? Inverse PCR?

“All four lines were fluorescence-sorted and screened at each generation. In the Aaeg-M line, a single recombination event was observed after 15 generations (>10,000 individuals screened in total).”

Data not shown?

“Of note, all tested recombinants in Aal-M appeared to be sterile.”

Can the authors spell out for the reader why this may be the case.

“In the competitiveness assay between”, “in the flight test”

The results section is written in a minimalist style listing the outcomes only. Can the authors very briefly summarize what each assay measures and how.

“We estimate that an operational speed for sex sorting would be around 60 larvae/sec (which corresponds to 200 larvae/mL in the reservoir water with our instrument settings) with a mean recovery rate of $69.6 \pm 1.1\%$ and no female contamination detected among more than 6,500 sorted males”

Can the authors better explain how this trade-off was arrived at as the best compromise.

“what is currently being done in genetic control field trials”

This needs a reference or a description.

Figures

Figure 1.

“M-linkage was obtained by piggyBac homing near the M-locus,”

Better not to use the term homing as this has a well-defined meaning. The mechanism here is unknown so it should be treated as a bias that is not currently explained.

Panels c,d,g,h:

X and Y axis are missing units. Are these panels showing actual data of exemplary sortings or are they made up? It would be good to provide more information on what we are seeing here.

Panel e and also in the main text. The exact way in which Nix YFP and Nix GFP plasmids were used/combined here isn't clear.

Figure 2.

Sorting panels miss units/scales/labels

Panel f: There seems to be some individuals in albopictus that fall in between clearly GFP positive and negative populations. Can the authors speculate why that is? Have those ever been looked at in terms of sex?

Figure 3a,e also c, g

This is a somewhat strange presentation. If I get this right every coloured dot represents a single individual that is either male or female? But the authors plot that on a graph that shows the percentage of males. Doesn't make a lot of sense. It would say that males are 100% percent male and females 0% male and hence the mean of all datapoints gives us a 50% sex ratio. I feel this very simple data is presented in an overly complicated manner. Same for the hatching in panels c & g. Overall, it seems more correct to just plot the means of the replicates and the sample size as done in panels d & h.

I don't feel strongly about it.

Figure 4a,b,c,d

Same as above, plotting just the mean would make this better. Again, the panels show that transgenic individuals are 100% on the "transgenic progeny axis" whereas wild types are 0%. Percentage can be plotted only for aggregates.

Reviewer #2 (Remarks to the Author):

In their manuscript entitled "Combining two Genetic Sexing Strains allows sorting of non-transgenic males for *Aedes* genetic control," Lutrat and colleagues describe a comprehensive system for identifying and sorting transgenic male and female mosquitoes using four novel sexing strains, one for male and female *Ae. aegypti* and *albopictus*, respectively, taking advantage of a small and conserved male-determine ('M') locus in each species. They further demonstrate the use of COPAS sorting for identifying and selecting labeled animals.

These reagents and the associated methodology suggests an accurate, efficient and cost-effective method for generating mass quantities of male mosquitoes for SIT and other applications and is a useful addition to the field of mosquito genetics and control. In addition, it provides new data on piggyBac homing techniques in *Aedes albopictus* which may be widely useful for the field. The fitness of these strains was catalogued in a systematic way that suggests they would be competitive, at least at a gross motor and life history level, if utilized for SIT. All told, this manuscript represents an important step forward that will be an excellent publication if some minor concerns can be addressed. The detailed methods and availability of statistical analyses and sequences of molecular reagents is to be applauded.

Major comments:

The up to 0.1% recombination rate of the Aa1-m strain (meaning 1 in 1000 'contamination' of sorted males by females) is potentially troubling if it cannot be detected or mitigated. The generation of a balancer chromosome (Fig 5h) is a tantalizing idea, but would need to be weighed against the potential (likely?) fitness costs of a large chromosomal inversion and, to date, does not exist. I find that the authors discuss these ideas appropriately but in scattered places in the manuscript, potentially understating the ramifications for use in actual release settings. Given the focus of this paper on generating a cost-effective deployment solution, I think these issues need to be discussed explicitly, even if the current 'state-of-the-art' suffers from much higher potential error rates, and potentially a 'middle ground' could be proposed in which COPAS sorting generates males which are then manually sorted at adult or pupal stages to ensure as close to zero tolerance as possible.

Minor comments:

The presumed non-recombining region surrounding the 'M' locus varies quite a bit depending on strain

(see ref 27,

Figure 1: I find this figure a bit busy, with many graphical elements (e.g. microscope for embryo injections) that may be unnecessary. A redesigned and streamlined figure may help the reader follow the logic of each experiment. In addition, some details that may be important (e.g. the distance between the M/m-locus and AAEL019619) would be a helpful addition to this graphical-abstract style figure.

Figure 3: I find the graphs difficult to read due in part to the large size of the diagrams above each panel. In addition, some instances of text being placed inconsistently with respect to the graphs (e.g. the significance indicators in Fig. 3g

Reviewer #3 (Remarks to the Author):

This well-written methods paper describes the generation of sex-locus marked strains in *Aedes* species as an approach for more efficient mosquito population replacement to control vector-borne disease. The method described is designed to supplement existing SIT or lethal transgene approaches. Though the sex locus in *Aedes albopictus* has not been thoroughly described, the authors have previously published evidence of *Albopictus*-specific Nix function as a male determining trait, which justifies moving forward with this approach in the current project. The authors used different methods to introduce marked M- or m-linked transgenes into *Aeeg* or *Aal*. They explained the number of generations that were checked and also determined the recombination rates for loss of the transgenes. Presumably, further optimization of this approach and/or detection methods could reduce loss of the transgene even further.

The major findings were that COPAS sorting errors were very rare but were higher than for the Verily multi-step sorter. Importantly, sorting issues associated with COPAS or other technical issues could perhaps be reduced under future iterations or with the use of other fluorescent sorters. In addition, the proposed method described here obviates the need for expensive automation equipment associated with Verily's system. Lastly, the developed GSS strains could also be used in other contexts, for example, in the study of the causes of sex distortion or other basic biological questions.

The authors go on to show that sex ratio distortion is not occurring in the transgenic lines. They also performed important experiments to show that fecundity and survival were not compromised in the transgenic strains.

Ln 39- Please italicize *Aedes* spp.

Ln 108- Please define "CS", as in *Aeeg*-CS.

Figures- the text in some of the figures is really too small to be practical. Please reconfigure them and increase font sizes of legends to enhance readability. For example, in Fig3A, the sample numbers could be moved to the figure legend.

Ln 169- The authors state, "Aal-M showed a 0.59 competitiveness compared to BiA (WT) and a similar competitiveness compared to Aal-CS (Figure 4b), which means that both Aal-M and Aal-CS had a reduced competitiveness as compared to WT ", however, Fig 4B indicates no significant difference in male competitiveness in Aal-CS vs WT. Please explain.

Ln 172- Please explain what is meant by 'escape rates' here or in the appropriate methods section.

Lns 259-273- Also, please explain in more detail how GSS-CS differs from GSS. They were compared extensively in this paragraph, however more context would help the reader follow along.

Lns 324-332- The authors describe a crossing strategy to generate non-transgenic mosquitoes. What would the purpose of such an approach be in practice? Please add an explanation of the use and benefits of this approach for those not immersed in this methodology.

Ln 340- Please explain the advantage of releasing non-transgenic males for readers who may not be familiar with SIT.

Ln 528- Please define PAM.

Ln 646- Were biological or technical replicates used for fitness analyses?

Ln 667- Please give more detail about the flight test device.

Point-by-point response to reviewers

Reviewers' comments:

Reviewer #1 (Remarks to the Author):

Lutrat et al. describe a set of transgenic *Aedes aegypti* and *Aedes albopictus* transgenic strains that, via linkage of fluorescent makers to either the M or m loci, allow using crossing and fluorescent larval sorting to generate transgenic or non-transgenic single sex mosquito populations. Such populations are the foundation for genetic control strategies based on sterility and the authors assess the limitations and cost-effectiveness of their approach compared to other strategies.

This is an impressive piece of work and clearly the cumulation of many years of mosquito wrangling. I appreciated that the authors tried to make this large collection of strains, data and results accessible via great figures and that they provide data and code that allows the reader to evaluate their findings and statistical models.

Because the paper summarizes a large set of experiments where events were screened for over several generations and in different ways it is sometimes descriptive and glances over some of the steps in describing how certain things were done obtained. I find this acceptable in this case as the huge amount of work summarized here is a tour de force of *Aedes* transgenesis. Nevertheless, I made some suggestions below to make the paper clearer and more accessible.

I could not fully evaluate the economic model presented by the authors as this is beyond my expertise but found that part of the paper among the most interesting.

Overall, the paper can be accepted with minor revisions, it is an impressive piece of work that will find its place in the genetic control literature.

Specific suggestions:

Abstract

“error-prone, and wasteful sex separation requirement” Requirements are not wasteful or error-prone. Particular methods are.	Thank you for spotting this. We replaced it by: “The Sterile Insect Technique is a valuable alternative but is limited by slow, error-prone, and wasteful sex-separation methods.”
“Scaling-up would allow the sorting of 100,000 first instar male larvae in under 1.5 hours with an estimated 0.01-0.1% female contamination.” This sentence doesn’t make sense. Scaling-up can always go further. If you refer to particular numbers of insects achievable you need to refer to specifics setups (x number of COPAS machines etc) when talking about scaling up. But that is best not done in the abstract.	Thank you for this remark. What we meant by scaling up was scaling up the mosquito rearing, as these are the figures that can be obtained on a single device. We clarified this sentence as follows: “In a mass-rearing facility, 100,000 first instar male larvae could be sorted in under 1.5 hours with an estimated 0.01-0.1% female contamination on a single machine.”

Results

“targeting the AAEL019619” Perhaps a short description of this gene and the rationale for targeting it is warranted.	We added more details: “In Ae. aegypti, linkage of an eGFP marker transgene to the m and M loci was achieved by CRISPR-Cas9 knock-in targeting a mucin-3A gene, AAEL019619, that was predicted to be central to the non-recombining region encompassing the sex-loci by Fontaine and colleagues (30)”
“stimulated by sequences derived from the Nix gene and cloned in the piggyBac transgenesis plasmid” Perhaps a little more description is warranted here.	To answer this comment and in accordance with your suggestion related to Fig. 1 we have reworded to avoid confusion with the homing observed in case of gene drive. The term "transposon homing" had been historically defined in Drosophila, which we refer here: “In Ae. albopictus, M-linkage of fluorescence markers was achieved by piggyBac preferential insertions near the masculinization gene, Nix, stimulated by the inclusion of Nix-derived sequences in the piggyBac transgenesis plasmid. A similar phenomenon, termed transposon homing, has been previously observed for P elements in Drosophila”
“Sequencing revealed that the transposon had landed” What method was used here? Inverse PCR?	We could not get long enough sequences in inverse PCR to map them specifically in the genome given its repeatedness. We used a nanopore sequencing method adapted from Gilpatrick et al. as described in the Methods section. We added a reference to the Methods paragraph in the main text so that readers can check it.
“All four lines were fluorescence-sorted and screened at each generation. In the Aaeg-M line, a single recombination event was observed after 15 generations (>10,000 individuals screened in total).” Data not shown?	Sexes in early generations were systematically COPAS-separated and verified later by careful visual examination of the female and male pupal batches + adult cages to search for mosquitoes of the wrong sex. None were found at early generations of the Aaeg-M strain. However, we did not keep record of the exact number of pupae screened at each generation. More sporadic COPAS sortings were then carried out in subsequent generations, and in the 15th one non-fluorescent male was visually detected among the females, indicative of a recombination event. This observation was recorded as a note but no photographs were taken, hence we added the word “visually” in our statement "In the Aaeg-M line, a single recombination event was visually recorded after 15 generations (>10,000 individuals screened in total)”

“Of note, all tested recombinants in Aal-M appeared to be sterile.” Can the authors spell out for the reader why this may be the case.	Since the first submission of this manuscript, we had the opportunity to perform an additional cross of WT females to a few recombinant, non-fluorescent Aal-M males, which this time proved fertile. Therefore, we removed the sterility statement. We don’t rule out, however, that some of these males can be sterile, which could be explained by recombination placing endogenous Nix in cis of m-linked genetic factors antagonistic to male fertility.
“In the competitiveness assay between”, “in the flight test” The results section is written in a minimalist style listing the outcomes only. Can the authors very briefly summarize what each assay measures and how.	We added the following sentences: “We tested the competitiveness of transgenic males by placing 30 transgenic males and 30 WT males in a cage with 30 virgin WT females and measuring the percentage of transgenic progeny in their offspring.” “Male flight ability was assessed through a standardized test consisting in placing 100 males in a small cup at the bottom of vertical tubes topped by a fan and counting how many males manage to escape through the tubes”
“We estimate that an operational speed for sex sorting would be around 60 larvae/sec (which corresponds to 200 larvae/mL in the reservoir water with our instrument settings) with a mean recovery rate of $69.6 \pm 1.1\%$ and no female contamination detected among more than 6,500 sorted males” Can the authors better explain how this trade-off was arrived at as the best compromise.	As shown on Fig. 5a, recovery decreases with speed, meaning that we had to accept a trade-off between rearing and producing extra mosquitoes because of the recovery rate and sorting fast enough for producing millions of mosquitoes weekly. $\geq 70\%$ recovery seemed to be necessary. Additionally, the contamination rate increases with speed (even though with 10k larvae we only observed mistakes at 300 larvae/sec, it is likely that with higher numbers we might find mistakes at lower speeds). We rephrased using “proposed” instead of “estimate” as no proper cost estimate was performed here.
“what is currently being done in genetic control field trials” This needs a reference or a description.	We added the following reference: “Guidance framework for testing the sterile insect technique as a vector control tool against aedes-borne diseases” World Health Organization and The International Atomic Energy Agency, 2020

Figures

Figure 1.

“M-linkage was obtained by piggyBac homing near the M-locus,”	Replaced by “In Ae. albopictus, M-linkage was obtained by piggyBac preferential integration
--	--

Better not to use the term homing as this has a well-defined meaning. The mechanism here is unknown so it should be treated as a bias that is not currently explained.	near the M-locus stimulated by the presence of Nix sequences” in the legend and “piggyBac preferential integration” in the Figure.
Panels c,d,g,h: X and Y axis are missing units. Are this panels showing actual data of exemplary sortings or are they made up? It would be good to provide more information on what we are seeing here.	The COPAS output is in arbitrary fluorescence units, hence the scale on Figures 1 and 2. All COPAS graphs are plotted from actual data for these strains. We added the following statements: “Both lines allow sex separation using a COPAS flow cytometer, as shown on the graphs generated from representative sorting data for these lines (arbitrary fluorescence units).” “Both Aal-M and Aal-m allow sex separation using a COPAS flow cytometer (representative sorting from actual data shown, in COPAS arbitrary fluorescence units).”
Panel e and also in the main text. The exact way in which Nix YFP and Nix GFP plasmids were used/combined here isn’t clear.	They have not been combined, several fluorochrome-bearing plasmids were injected at the same time, generating individuals carrying more than one transgene. First, we backcrossed the transgenic males to isolate the different insertions, then we followed the insertions of interest individually until we selected the YFP one because it showed the best COPAS sex separation pattern. GFP was shown on the 1st step of the figure as an example, but it may be confusing as we select YFP in the end, hence we changed this part of the figure.

Figure 2.

Sorting panels miss units/scales/labels	These are direct outputs from the COPAS device, they are always without scales with just the Lg(fluorochrome) labels. We miss the raw data for one of the replicates therefore we could not plot them as presented in figure 1. We added the following sentence to the figure legend: “No scales are displayed by the software as it uses arbitrary fluorescence units.”
Panel f: There seems to be some individuals in albopictus that fall in between clearly GFP positive and negative populations. Can the authors speculate why that is? Have those ever been looked at in terms of sex?	Yes, we sorted these individuals separately and observed that they were either dead male larvae or male larvae with lower GFP expression at the time of the sorting. We added the following statement to the figure legend: “ Larvae of intermediate fluorescence were observed to be either dead fluorescent larvae or larvae with lower GFP expression at the time of the sorting.

	All the individuals that reached the pupal stage were confirmed to be males.”
--	---

Figure 3a,e also c, g

This is a somewhat strange presentation. If I get this right every coloured dot represents a single individual that is either male or female? But the authors plot that on a graph that shows the percentage of males. Doesn't make a lot of sense. It would say that males are 100% percent male and females 0% male and hence the mean of all datapoints gives us a 50% sex ratio. I feel this very simple data is presented in an overly complicated manner. Same for the hatching in panels c & g. Overall, it seems more correct to just plot the means of the replicates and the sample size as done in panels d & h.

I don't feel strongly about it.

=> The reason why we plotted it this way was because our statistical test uses a binomial distribution that takes into account both the number of individuals in each replicate and the number of replicates, hence we wanted to make both figures. The binary dots referred to the right y-axis, stating "Males" versus "Females". However, we admit that, given the size of the final graph, it makes it somewhat busy, plus the full data are already present in the supplementary. Therefore we removed these data points from the graph.

Figure 4a,b,c,d

Same as above, plotting just the mean would make this better. Again, the panels show that transgenic individuals are 100% on the "transgenic progeny axis" whereas wild types are 0%. Percentage can be plotted only for aggregates.

=> We simplified this graph as well.

Reviewer #2 (Remarks to the Author):

In their manuscript entitled "Combining two Genetic Sexing Strains allows sorting of non-transgenic males for *Aedes* genetic control," Lutrat and colleagues describe a comprehensive system for identifying and sorting transgenic male and female mosquitoes using four novel sexing strains, one for male and female *Ae. aegypti* and *albopictus*, respectively, taking advantage of a small and conserved male-determine ('M') locus in each species. They further demonstrate the use of COPAS sorting for identifying and selecting labeled animals.

These reagents and the associated methodology suggests an accurate, efficient and cost-effective method for generating mass quantities of male mosquitoes for SIT and other applications and is a useful addition to the field of mosquito genetics and control. In addition, it provides new data on piggyBac homing techniques in *Aedes albopictus* which may be widely useful for the field. The fitness of these strains was catalogued in a systematic way that suggests they would be competitive, at least at a gross motor and life history level, if utilized for SIT. All told, this manuscript represents an important step forward that will be an excellent publication if some minor concerns can be addressed. The detailed methods and availability of statistical analyses and sequences of molecular reagents is to be applauded.

Major comments:

The up to 0.1% recombination rate of the Aa1-m strain (meaning 1 in 1000 ‘contamination’ of sorted males by females) is potentially troubling if it cannot be detected or mitigated. The generation of a balancer chromosome (Fig 5h) is a tantalizing idea, but would need to be weighed against the potential (likely?) fitness costs of a large chromosomal inversion and, to date, does not exist. I find that the authors discuss these ideas appropriately but in scattered places in the manuscript, potentially understating the ramifications for use in actual release settings. Given the focus of this paper on generating a cost-effective deployment solution, I think these issues need to be discussed explicitly, even if the current ‘state-of-the-art’ suffers from much higher potential error rates, and potentially a ‘middle ground’ could be proposed in which COPAS sorting generates males which are then manually sorted at adult or pupal stages to ensure as close to zero tolerance as possible.

Answer:

Thank you for your comment.

Concerning the recombination rates, we added the following paragraph to address this issue more clearly in the discussion: “Of note, up to 0.3% of contaminant females were considered acceptable in recent SIT-IIT assays while up to 1% is tolerated for SIT alone . Moreover, contaminant females would be subjected to the same irradiation dose as males, which has been proven to cause 100% sterility in females. Therefore, the main concern regarding these contamination rates is that contaminant females have to be removed from the colonies in the mass-rearing facilities so that they do not accumulate. This has been taken into account in our simulations by including filter colonies that would be sex-sorted as larvae and double-checked as pupae. In cases where closer to 0% contamination is required in the release batches, a quality control step using automated pupal sorters could also be implemented prior to adult emergence.”

To include the caveat that inversions lacking major fitness costs will need to be generated, we expanded the discussion with the following sentence: “However, inversions will need to be carefully analysed to ensure that they don’t introduce unwanted fitness costs into the sexing strain, which has been observed in other species.”

Minor comments:

The presumed non-recombining region surrounding the ‘M’ locus varies quite a bit depending on strain (see ref 27,	This is true. This is why, for the Ae. aegypti strains, we targeted a gene predicted to be very central to the non-recombining region. However, in light of these results, we now know that we should have targeted a gene even closer to the sex loci sensu stricto in order to obtain lower/no recombination, but the exact position of the m locus in females was unknown. Future work may refine this.
Figure 1: I find this figure a bit busy, with many graphical elements (e.g. microscope for embryo injections) that may be unnecessary. A redesigned and streamlined figure may help the	Thank you for these suggestions. We redesigned it a bit, hoping it is now clearer.

reader follow the logic of each experiment. In addition, some details that may be important (e.g. the distance between the M/m-locus and AAEL019619 would be a helpful addition to this graphical-abstract style figure.	
Figure 3: I find the graphs difficult to read due in part to the large size of the diagrams above each panel. In addition, some instances of text being placed inconsistently with respect to the graphs (e.g. the significance indicators in Fig. 3g	We edited this figure in line with your comments and that of the other reviewers.

Reviewer #3 (Remarks to the Author):

This well-written methods paper describes the generation of sex-locus marked strains in *Aedes* species as an approach for more efficient mosquito population replacement to control vector-borne disease. The method described is designed to supplement existing SIT or lethal transgene approaches. Though the sex locus in *Aedes albopictus* has not been thoroughly described, the authors have previously published evidence of *Albopictus*-specific Nix function as a male determining trait, which justifies moving forward with this approach in the current project. The authors used different methods to introduce marked M- or m-linked transgenes into *Aeeg* or *Aal*. They explained the number of generations that were checked and also determined the recombination rates for loss of the transgenes. Presumably, further optimization of this approach and/or detection methods could reduce loss of the transgene even further.

The major findings were that COPAS sorting errors were very rare but were higher than for the Verily multi-step sorter. Importantly, sorting issues associated with COPAS or other technical issues could perhaps be reduced under future iterations or with the use of other fluorescent sorters. In addition, the proposed method described here obviates the need for expensive automation equipment associated with Verily's system. Lastly, the developed GSS strains could also be used in other contexts, for example, in the study of the causes of sex distortion or other basic biological questions.

The authors go on to show that sex ratio distortion is not occurring in the transgenic lines. They also performed important experiments to show that fecundity and survival were not compromised in the transgenic strains.

Ln 39- Please italicize Aedes spp.	Oops, edited.
Ln 108- Please define "CS", as in Aeeg -CS.	We added the following precision: "Their progeny (Aeeg -CS, 'CS' standing for 'Crossing Scheme') comprised ..."
Figures- the text in some of the figures is really	We edited the figures, as we did not realize they

too small to be practical. Please reconfigure them and increase font sizes of legends to enhance readability. For example, in Fig3A, the sample numbers could be moved to the figure legend.	would be hard to read at this scale. Following recommendations from the other reviewers as well, we removed some elements which freed space for increasing font sizes.
Ln 169- The authors state, “Aal-M showed a 0.59 competitiveness compared to BiA (WT) and a similar competitiveness compared to Aal-CS (Figure 4b), which means that both Aal-M and Aal-CS had a reduced competitiveness as compared to WT “, however, Fig 4B indicates no significant difference in male competitiveness in Aal-CS vs WT. Please explain.	We measured competitiveness by counting the proportion of transgenic progeny in a competition assay between transgenic and non transgenic males. Aal-CS and WT being both non-transgenic, direct comparison was not possible this way. In figure 4b, we show a competition assay between Aal-CS (non-transgenic) and Aal-M (transgenic) males. In this assay, no significant difference was observed, meaning that the competitiveness of Aal-CS was comparable to that of Aal-M males. Aal-M males being less competitive than WT (fig. 4a), we concluded that Aal-CS were, too.
Ln 172- Please explain what is meant by ‘escape rates’ here or in the appropriate methods section.	We added the following sentence: “Male flight ability was assessed through a standardized test consisting in placing 100 males in a small cup at the bottom of vertical tubes topped by a fan and counting how many males manage to escape through the tubes”.
Lns 259-273- Also, please explain in more detail how GSS-CS differs from GSS. They were compared extensively in this paragraph, however more context would help the reader follow along.	We added the following sentence: “We considered the case of GSS-CS (to produce non-transgenic males) separately from that of GSS as it requires two rearing colonies (one for each strain to be used) and two filter colonies instead of one (see the ‘Mass rearing processes’ schemes in Supplementary Data 3), which translate into more mosquitoes to be reared and requiring extra space and equipment.”
Lns 324-332- The authors describe an crossing strategy to generate non-transgenic mosquitoes. What would the purpose of such an approach be in practice? Please add an explanation of the use and benefits of this approach for those not immersed in this methodology.	We added the following precision: “This strategy could be particularly useful in countries where genetically modified insects are not strictly banned but where releasing transgenic mosquitoes might cause strong public concern.”
Ln 340- Please explain the advantage of releasing non-transgenic males for readers who may not be familiar with SIT.	Some of the discussion paragraphs got mixed, we reordered them for it to be easier to follow. Given the explanation added above and the ones that come right after, we believe that the benefit of producing non-transgenic males to better meet local requirements is now clear
Ln 528- Please define PAM.	We added the following precision: “(...) brackets indicate the Protospacer Adjacent Motif, PAM”
Ln 646- Were biological or technical replicates used for fitness analyses?	They were biological replicates. The precision was added to the methods section.

Ln 667- Please give more detail about the flight test device.

We added the following precision: “Pools of males were placed in a tight cup at the bottom of several vertical tubes topped by a fan. In order to escape the tight cup, they have to flight up the vertical tubes against the airflow caused by the fan. After two hours, the number of males that escaped versus the number that did not were counted and escape rates were compared between strains.”

Updated figures:

Graphical abstract

Automated sorting of transgenic or non-transgenic males

Changes:

Replaced 'piggyBac homing' by 'PiggyBac "guided"'

Figure 1

Changes:

Added distance between M/m-locus and AAEL19619 on panels a) and b). Corrected fluorochrome on panel c). Removed microinjection schemes. Increased font size. Changed text in panels c) and d). Edited style for better readability.

Figure 2

Unchanged

Figure 3

Changes:

Removed individual datapoints and secondary y-axes. Increased font sizes. Reduced size of explanatory drawings. Corrected statistics in panels b and f (hurdle model, see Suppl. Data 2). Repositioned statistics on graphs.

Figure 4

Changes:

Removed individual datapoints and secondary y-axes. Increased font sizes. Moved explanatory drawings to the left. Repositioned statistics on graphs.

Figure 5

Unchanged

REVIEWERS' COMMENTS:

Reviewer #1 (Remarks to the Author):

The authors have adressed all my comments and the manuscript has been given another layer of polish. It's a great paper and it's ready to go.

Reviewer #2 (Remarks to the Author):

I have now re-read the revised manuscript and replies to the reviewers, and I am satisfied that they have addressed all points raised. I support the publication of this manuscript in its present form.

Reviewer #3 (Remarks to the Author):

The authors have adequately addressed the reviewer concerns.